# Associations between the Gut Microbiota, Race, and Ethnicity of Patients with Colorectal Cancer: A Pilot and Feasibility Study

**DOI:** 10.3390/cancers15184546

**Published:** 2023-09-13

**Authors:** Sorbarikor Piawah, Than S. Kyaw, Kai Trepka, Anita L. Stewart, Rosa V. Mora, Dalila Stanfield, Kendall Levine, Erin L. Van Blarigan, Alan Venook, Peter J. Turnbaugh, Tung Nguyen, Chloe E. Atreya

**Affiliations:** 1Department of Medicine, University of California, San Francisco, CA 94143, USA; 2Helen Diller Family Comprehensive Cancer Center, San Francisco, CA 94143, USA; 3UCSF Center for Aging in Diverse Communities, San Francisco, CA 94143, USA; 4Department of Microbiology and Immunology, University of California, San Francisco, CA 92521, USA; 5School of Medicine, University of California, San Francisco, CA 92521, USA; 6Institute for Health & Aging, University of California, San Francisco, CA 92521, USA; 7School of Nursing, University of California, San Francisco, CA 92521, USA; 8Zuckerberg San Francisco General Hospital, San Francisco, CA 94110, USA; 9Department of Urology, University of California, San Francisco, CA 92521, USA; 10Department of Epidemiology and Biostatistics, University of California, San Francisco, CA 92521, USA; 11Chan Zuckerberg Biohub-San Francisco, San Francisco, CA 40385, USA; 12Osher Center for Integrative Medicine, San Francisco, CA 94115, USA

**Keywords:** colorectal cancer, microbiome, disparities, recruitment, implementation, race

## Abstract

**Simple Summary:**

The trillions of microorganisms found in the human gut have broad impacts on health and disease including altering the risk of colorectal cancer. Given the emerging evidence that both the microbiome and colorectal cancer differ between racial and ethnic groups, we sought to explore the relationship between the gut microbiome and cancer health disparities. We performed a pilot and feasibility study in which 30 colorectal cancer patients from different racial and ethnic backgrounds were recruited to provide a stool sample and to complete a dietary survey. The majority (18/30) of participants were successfully sampled, providing preliminary support for associations between their gut microbes according to racial and ethnic background. Overall, this study supports the feasibility of analyzing the links between gut microbes and colorectal cancer disparities, laying the groundwork for large follow-up studies.

**Abstract:**

Background: Colorectal cancer (CRC) is more prevalent among some racial and ethnic minority and low socioeconomic status populations. Although the gut microbiota is a risk factor for CRC and varies with race and ethnicity, its role in CRC disparities remains poorly understood. Methods: We examined the feasibility of recruiting sociodemographically diverse CRC patients for a microbiome study involving a home stool collection. We also explored whether race and ethnicity were associated with gut microbiome composition. We recruited Black/African American, Hispanic/Latino, and non-Hispanic White patients who were receiving care for active CRC to complete a comprehensive dietary and lifestyle survey, self-collect a stool sample, and complete an exit interview. Gut microbial diversity and composition were analyzed using 16S rRNA gene sequencing. Results: 30 individuals consented (of 35 who were eligible and contacted) with 5 (17%) Black/African American, 11 (37%) Hispanic/Latino, and 14 (46%) non-Hispanic White. A total of 22 (73%) completed the dietary and lifestyle survey; 18 (63%) returned a stool sample. Even after controlling for socioeconomic, dietary, or treatment-related covariates, microbiome composition was associated with race and ethnicity. Fusobacteriota (a phylum associated with the development and progression of CRC) was significantly higher in the Black/African American group compared to others, and microbial diversity was higher in samples from non-Hispanic White individuals compared to Hispanic/Latino individuals. Conclusion: Our study shows that it is feasible to recruit and collect stool samples from diverse individuals with CRC and found significant associations in gut microbial structure with race and ethnicity.

## 1. Introduction

Colorectal cancer (CRC) is a common and challenging disease [1]. Although CRC incidence has declined in the last 20 years, racial and ethnic disparities persist [2], with Black/African Americans 20% more likely to be diagnosed and 40% more likely to die than Non-Latino White individuals [3]. Rates of young-onset CRC (age less than 50 at diagnosis) are rising most rapidly among Hispanic/Latino Americans [4,5]. Past studies exploring these disparities have demonstrated the contribution of factors such as race and ethnic differences in access to care, diet, and lifestyle [6]. However, efforts to reduce disparities by targeting these factors alone have had limited success, suggesting the presence of other factors [7].

There is a growing literature linking the gut microbiota (the trillions of microbes in the gastrointestinal (GI) tract) and their aggregate genomes (the gut microbiome) in CRC risk and treatment [8,9,10,11,12,13,14,15,16,17,18]. Gut bacteria including *Fusobacterium nucleatum*, *Enterococcus faecalis*, *Bacteroides fragilis*, and *Streptococcus gallolyticus* may contribute to the pathogenesis of CRC through interactions with immune cells, initiating inflammatory and pro-oncogenic cascades, releasing genotoxins, and other mechanisms [8,9,10,11,12,13,14,15,16,17,18]. Similarly, the gut microbiome can play a role in the efficacy and toxicity of chemotherapy by altering the absorption and metabolism of drugs through direct and indirect methods [19,20]. Several studies have demonstrated the impact of the gut microbiome on the efficacy of immunotherapy, via interactions between gut bacteria, immune cells, and the tumor microenvironment [8,19].

However, there are still few studies exploring the interactions between CRC and gut microbiota with race and ethnicity. Outside the context of CRC, variability in the gut microbiota has been associated with race and ethnicity, but the mechanisms responsible are not yet well understood [21,22,23]. A few pilot studies have begun to describe differences in the gut microbiome of patients with CRC by race [24,25]. However, research has been limited by challenges in recruiting adequate numbers of diverse individuals to participate and complete the complex study procedures needed to accurately study these associations in a reproducible manner.

We conducted a pilot study to assess the feasibility of recruiting sociodemographically diverse CRC patients for a microbiome study involving a home stool collection and a diet and lifestyle questionnaire. Here, we present these feasibility data and the results of exploratory analyses of racial and ethnic differences in the microbiome composition of the samples. We also discuss suggestions for modifications to the methods to increase the chances of success in a larger-scale follow-on study.

## 2. Materials and Methods

### 2.1. Study Design

This was a prospective cross-sectional pilot study of patients with CRC receiving care at the Zuckerberg San Francisco General Hospital (ZSFGH), San Francisco’s largest safety net public hospital, or at the gastrointestinal oncology clinics at the University of California San Francisco (UCSF). This study was approved by the UCSF Institutional Review Board.

### 2.2. Eligibility

We aimed to recruit 60 adult (≥18 years of age) English-speaking patients self-identifying as Black/African American, Hispanic/Latino, or non-Hispanic White who were receiving active treatment for any stage of CRC. We limited our recruitment to these racial and ethnic groups in part because disparities in CRC incidence are most pronounced between them, and because of limitations on the budget for our one-year pilot.

Given the influence of cancer-directed therapies on the gut microbiome [8] and the expected duration of these effects, we excluded individuals who were unable to have a break of at least 12–14 days from chemotherapy, biologic therapy, or immunotherapy prior to stool collection. Similarly, given the potential impact of antibiotics on the composition of the gut microbiome [26,27], we also excluded those who had either received antibiotics within 4 weeks prior to enrollment, or completed a course of antibiotics longer than 2 weeks in the 6 months before enrollment.

### 2.3. Recruitment

We sought to enroll approximately equal numbers of Black/African American, Hispanic/Latino, and non-Hispanic White patients. We prioritized the recruitment and enrollment of Hispanic and Black/African American individuals early in the study period and planned to cap enrollment of non-Hispanic White individuals to no more than 1/3 of the study population.

To identify potentially eligible patients, we obtained a list from two sources: direct referrals of any patient with active CRC from treating physicians and weekly review of providers’ schedules at each site, identifying patients for whom colon or rectal cancer was listed as either the diagnosis or the reason for their visit. Potentially eligible patients from both sources were entered in a secure screening log including name, medical record number, diagnosis, race, and ethnicity.

The study team was racially and ethnically diverse and conducted a brief screen of the electronic health record (EHR) to assess eligibility prior to contacting the patient. Eligibility criteria were active disease on imaging, not receiving antibiotics, being able to have a 12–14-day break in therapy, and English fluency. Eligible patients were then contacted. Due to in-person research restrictions due to the COVID-19 pandemic during the study period, this initial contact was restricted to telephone calls at both sites. Coordinators were instructed to call potentially eligible individuals 3 times over the course of 4 weeks before considering them “unable to be reached”. Due to patient confidentiality concerns, voicemails were not left, and email was not utilized. Once a person was contacted, a formal telephone recruitment script was utilized at both study sites. Research staff briefly described the study including its aims, procedures, duration, and compensation (script details are included in Appendix A). Interested patients were then screened again for confirmation of eligibility (queried primarily about antibiotic use and current cancer therapy).

Eligible patients who completed the telephone interview were then invited to schedule a separate consent meeting either via telephone with documents signed electronically or in person at an upcoming clinical visit.

### 2.4. Study Procedures

We collected: (1) relevant medical history and eligibility criteria via review of EHR and at the initial telephone contact, (2) diet information via an online or paper comprehensive survey estimated to take 1 h, and (3) a stool biospecimen. At the end of the study, a semi-structured exit interview was administered via telephone or in person.

(1)Review of EHR: After consent, clinical and medical history were collected through review of participants’ EHR: clinical details of their CRC diagnosis (e.g., anatomic subsite, stage), treatment history, family history, antibiotic use, and other medical history not captured in the comprehensive survey. Demographics collected from the EHR included self-reported gender, age, and race and ethnicity.(2)Comprehensive survey: Usual dietary intake was measured using a comprehensive diet and lifestyle survey that incorporated a validated food frequency questionnaire (FFQ) developed at Harvard [28]. This FFQ is utilized in ongoing microbiome studies at UCSF and nationally. The survey also collected height and weight, smoking and drug history, alcohol use, sun exposure, sleep, physical activity, supplement use, and complementary medicines. Medical history was also queried, including family medical history, personal medical history, and early life history. Individuals who consented via telephone completed the survey electronically using a Research Electronic Data Capture (REDCap) database [29,30]. Those who completed consent in person were given the option of completing the survey in person on paper with the research coordinator or online in REDCap. The preferred method of survey completion was also recorded. (3)Biospecimen collection: Participants received a stool collection kit. Those who completed consent via telephone received the kit in the mail via FedEx while those who completed consent in person received the kit on the day of consent. Kits were designed by the Harvard Chan Microbiome Health Center BIOM-Mass program and have been used for other pilots at UCSF and Harvard [31]. Kits contained detailed instructions and all materials necessary for the collection of one stool sample, including the following:1 ethanol collection tube1 anaerobic collection tube2 bio-specimen bags1 pair of gloves2 paper accessories to attach to toilet seat2 spatulas4 barcode labels (2 for samples, 1 for the stool questionnaire and 1 extra)Pre-paid and pre-addressed FedEx envelopes

Briefly, participants were instructed to attach a stool collection paper accessory to a toilet seat and use a spatula to collect one scoop of stool each from the same bowel movement into the anaerobic culture tube and the ethanol tube. Samples collected from ostomy bags were permitted. If desired, the coordinator was available to review instructions with participants via telephone. Participants were also asked to complete a short questionnaire on paper at the time of sample collection to assess bowel habits and stool consistency (i.e., Bristol Stool Scale) and to provide an opportunity to give written feedback on the collection process. Lastly, participants were instructed to mail back the collected samples and survey to UCSF using pre-paid and pre-addressed FedEx envelopes. Upon receipt, samples were immediately banked and stored at −80 °C. 

Participants were contacted via telephone at least weekly for up to approximately 1 month for reminders to complete stool collection and return kits. A patient was considered lost to follow up if a kit was never returned and they were unable to be reached in this time.

(4)Exit interview to assess acceptability of study procedures:
Upon confirmation of receipt of collected samples in our microbiome lab, participants who returned the kit were contacted to complete an exit interview. Those who did not complete all study tasks were not contacted. The exit interview was conducted by research staff via telephone or in person, and elicited feedback on reasons for participation in the study, overall experience, and difficulties with the stool collection process. Participants were also given an opportunity to provide suggestions for improvement. Upon completion of this interview, participants received a $25 gift card.

### 2.5. Tracking Recruitment

To assess the feasibility of recruitment, we documented patient loss at each step and whether there were specific sources of losing potential patients throughout the recruitment process, as detailed in Appendix A. A secure Excel database was used to track patients identified from a review of provider schedules and referrals. The date of EHR review and reasons for ineligibility were recorded. For potentially eligible participants, we recorded the method and date of initial telephone contact, whether they were interested in the study, additional reasons for ineligibility, and reasons for declining the study. For those willing to consent, we recorded the completion of a consent visit and any further reasons for ineligibility. In addition, we report the time needed to meet our target sample size of 60.

### 2.6. Tracking Completion of Study Procedures

To assess the feasibility of data collection, we developed a detailed data collection protocol, including strategies for assuring completion of all assessments and methods for tracking compliance with the assessments. Upon receipt of consent documents, participants were registered in the REDCap database. REDCap was also used to track each step from participant consent to completion of the exit interview, including completion rates of the diet/lifestyle survey, stool kit survey, and the exit interview, as well as completion of stool collection, and reasons for non-completion of study components. We report the proportion of consented participants who completed stool collection, and the proportion of consented participants who completed the survey.

### 2.7. Data Analysis

We used descriptive statistics [means (standard deviation, SD) and median (interquartile range, IQR) for continuous variables and proportions for categorical variables] to summarize participant characteristics at enrollment.

### 2.8. Microbiome Analysis

DNA was extracted from ethanol-preserved stool samples using the International Human Microbiome Standard operating procedure (IHMS Protocol Q) [32] and the 16S rRNA gene sequencing library was constructed using dual error-correcting barcodes. Briefly, quantitative primary PCR was performed using KAPA HiFi Hot Start kit (KAPA KK2502) and V4 515F/806R Nextera primers. The amplified products were diluted 1:100 in UltraPure DNase/RNase-free water and were barcoded using unique dual indexing primers. The products were quantified using Quant-iT PicoGreen dsDNA Assay Kit (Invitrogen P11496) and pooled at equimolar concentrations. The pooled library was quantified via qPCR using the KAPA Library Quantification Kit (KAPA KK4824) and sequenced on the Illumina MiSeq platform. Demultiplexed sequences were processed using a 16S rRNA gene analysis pipeline [33]. High-quality reads were then analyzed using qiime2R [34] and phyloseq [35] in R. Alpha diversity was assessed through a number of observed amplicon sequence variants (ASVs) subsampled at 8166 reads and Shannon diversity index. Beta diversity was analyzed using the Bray–Curtis metric. For association analysis, education was binned into yes/no post-secondary degree (including Associate’s, Trade/Vocational, Bachelor’s, and Doctoral) and diet-related variables were quantitated as described previously [36], with American Cancer Society (ACS) diet sub-scores used to score the intake of whole grains, fruit/vegetables, meat, alcohol, and BMI. Analysis of variance (ANOVA) was used to test associations between covariates and Shannon alpha diversity. Multivariate permutational analysis of variance (PERMANOVA) on the Bray–Curtis distance matrix was used to uncover the associations between covariates and microbial community structures. A term-wise PERMANOVA with race and ethnicity as the final term (distance ~ Σcovariate + race and ethnicity) was used to test the effect of dietary, socioeconomic, and treatment-related confounders on the relationship between race and ethnicity and microbial community structure. For association analyses, nominal *p* < 0.05 was considered significant. Differentially abundant ASVs were determined using DESeq2 package in R with significance defined as |log_2_ fold-change| > 1 and false discovery rate (FDR) < 0.1. Sequencing data have been deposited under the NCBI BioProject PRJNA918703.

### 2.9. Thematic Analysis of Exit Interviews

Interviews were transcribed in real time by the interviewer. Briefly, themes from open-ended questions were identified through inductive coding of each interview transcript by two separate blinded reviewers: one used Dedoose software, whereas another coded by hand. The principal investigator served as a third reviewer to confirm emerging themes. Interviews also included two Likert scale questions that were analyzed using descriptive statistics.

## 3. Results

### 3.1. Recruitment

As described in Figure 1, during the 1-year study period, we received 302 patient names, a majority from a review of providers’ schedules across both sites—278 from UCSF and 24 from ZSFGH. Self-identified race and ethnicity were as follows: 31 Black/African American (10%), 61 Hispanic/Latino (20%), and 210 non-Hispanic White (70%).

Of the 302 patient charts reviewed, 177 (59%) across both sites (17 at ZSFGH, 160 at UCSF) were ineligible. The reasons for ineligibility were as follows: use of antibiotics outside of eligibility windows (11%), no evidence of active disease or on surveillance (32%), ineligible concurrent treatment (16%), preferred language other than English (6%), provider discretion (9%), and other reasons (26%). Other reasons included patients who were deceased, insufficient records to confirm eligibility, those who were enrolled in another study, and those who did not have histologic confirmation of CRC. 

In total, 125 participants were potentially eligible (7 at ZSFGH and 118 at UCSF). We made initial telephone contact with all 7 eligible participants at ZSFGH and 45 eligible participants at UCSF (a total of 52, 42% of 125 potentially eligible). The remainder (73) were unable to be reached despite three phone calls during business hours on weekdays, over 4 weeks. 

Of the 52 patients for whom initial telephone contact was made, confirmation of eligibility via self-report resulted in 9 becoming ineligible due to new use of antibiotics, newly on surveillance, clinical decline, or change in therapy. Of the 43 eligible, 8 (19%) declined participation: 5 due to lack of interest in the study, 2 due to feeling overwhelmed by their disease, and 1 due to irregular bowel habits making it difficult to collect stool. The remaining 35 of these 52 individuals (67%) who were eligible and interested in the study agreed to be contacted again for formal consent. 

Of the 35 who agreed to be re-contacted for consent, 5 (11%) did not return repeat phone calls to schedule a formal consent visit. The remaining 30 consented to participate in the study. A total of 4 individuals were recruited from ZSFGH and consented in person in a clinic, whereas the 26 individuals recruited from UCSF consented via telephone.

### 3.2. Participation Rate by Race and Ethnicity

Of the sampling frame of 302 names, 52 individuals were eligible and able to be reached to introduce the study, and 30 (58%) ultimately consented. Characteristics for enrolled patients are presented in Table 1. The participation rate was 42% (5/12) among Black/African American individuals, 55% (11/20) among Hispanic/Latino individuals, and 70% (14/20) among non-Hispanic White individuals.

### 3.3. Completion of Study Procedures

For the 30 individuals enrolled, Figure 1 presents the completion rates of study procedures. Of the 30 individuals who consented to the study, 1 withdrew from the study immediately after consenting due to a synchronous diagnosis of breast cancer. 

Of the 30 who consented, 22 (73%) completed the comprehensive survey—18 completed this survey online, whereas 4 individuals receiving care at ZSFGH completed it on paper with the assistance of a research coordinator. One became ineligible after consent but before receiving the collection kit and survey materials. Two individuals passed away between consent and completion of surveys, two were ineligible due to the new use of antibiotics, and three were provided a survey link but never completed and were lost to follow-up despite three reminder phone calls. All sections of returned surveys were completed. After receiving the kit, seven participants withdrew. All 18 remaining participants (60%) returned the stool kits. 

There were no stool kits returned without completion of the survey. A total of 17 of those who returned stool kits also completed an exit interview, while 1 was lost to follow-up. Of these 17 individuals, 15 (88%) were from UCSF and 2 (12%) from ZSFGH. Four (24%) identified as Black/African American, six (35%) as Hispanic/Latino, and seven (41%) identified as non-Hispanic White.

### 3.4. Retention Rate by Race and Ethnicity

The overall retention rate (proportion of individuals who completed all study tasks) was 61% (17 of 28 consented individuals who were alive at the end of the study period); 100% (4/4) among Black/African American individuals, 60% (6/10) among Hispanic/Latino individuals, and 50% (7/14) among non-Hispanic White individuals.

### 3.5. Acceptability of Study Procedures

All participants in the exit interview stated that they were likely or very likely to collect stool at home if requested by their doctor as part of clinical care. A total of 94% were likely or very likely to participate in another study that requires home stool collection using the same kit, 88% were likely or very likely to participate in another study that requires completion of the same comprehensive survey, and 65% were likely or very likely to recommend the study to a friend.

Thematic analysis demonstrated that at least half of the participants interviewed had no problems with the study procedures. Most described the sample collection process and survey as easy and chose to participate in the study to help other patients with cancer, and to contribute to the field of microbiome research. Below are some suggestions raised by participants.

#### 3.5.1. Experience with Stool Collection Process

Six participants felt that the collection process was messy and complicated as the collection tubes were too small and the toilet seat cover was difficult to place. Four felt that the instructions included in the kits were not easy to follow and contained a lot of information. The most frequent suggestions were to include larger containers for collection and to simplify the instructions. One person suggested including a video demonstration.

#### 3.5.2. Experience with the Comprehensive Survey

The most common theme was that the survey was too long and time-consuming, and that it was difficult to remember all foods eaten. Two participants suggested that participants should be reminded to keep a food diary to better recall foods eaten prior to completion of the survey whereas three suggested shortening the survey. 

### 3.6. Results of Microbiota Analyses

Despite the small sample size of this pilot and feasibility study relative to prior work on the gut microbiota with respect to race and ethnicity [22,23], we found statistically significant differences between groups. All 18 samples collected from Black/African American (AA), Hispanic/Latino (L), and non-Hispanic White (W) participants were evaluable and successfully sequenced, with comparable sequencing depth between groups (Appendix A).

Race and ethnicity were significantly associated with overall gut microbial community structure, accounting for 15% of the variation in our dataset (Figure 2a). We did not find any significant association with reported biological sex (Figure 2a). Principal coordinates analysis showed a clear separation between the microbiotas of Hispanic/Latino and non-Hispanic White individuals, with Black/African American individuals more dispersed (Figure 2a). Microbial diversity, assessed using the observed amplicon sequence variants (ASVs) and the Shannon diversity index, was significantly higher in samples from non-Hispanic White relative to Hispanic/Latino individuals (Figure 2b; Appendix A). Black/African American individuals also trended higher but did not reach statistical significance (Figure 2b; Appendix A).

We also detected significant differences in microbial abundances between racial and ethnic groups at the phylum and ASV levels (Figure 2c,d; Appendix A). The Hispanic/Latino group had significantly higher Actinobacteriota compared to the non-Hispanic White group; Black/African American individuals also trended lower relative to the Hispanic/Latino group (Figure 2c,d, Appendix A). Of note, Fusobacteriota, a phylum that contains *Fusobacterium nucleatum*, a species consistently associated with colorectal cancer pathogenesis [8], was significantly higher among Black/African American participants compared to non-Hispanic White participants (Figure 2d; Appendix A). 

We performed all possible pairwise comparisons to search for differentially abundant ASVs. Compared to Hispanic/Latino participants, Black/African American participants had 17 depleted ASVs and 18 enriched ASVs (Figure 2e, Appendix A). Compared to non-Hispanic White participants, Black/African American participants had 23 ASVs depleted and 6 ASVs enriched. Similarly, 33 ASVs were depleted and 12 ASVs were enriched among Hispanic/Latino participants compared to non-Hispanic White participants (Figure 2e, Appendix A). *Fusobacterium*, a genus associated with colorectal cancer, was only detected in 4/18 (22%) samples (Appendix A). 

Next, we set out to determine whether the relationship between race, ethnicity, and gut microbiota might be confounded by socioeconomic, dietary, and/or treatment (prior bowel surgery, radiation, and presence of any ostomy or ileostomy) covariates. None of the tested covariates were significantly associated with microbial diversity; however, we were able to confirm the association between microbial diversity and race and ethnicity (Appendix A). Similar results were observed when comparing gut microbial community structure; however, one additional covariate (prior ileostomy) reached statistical significance (Appendix A). Finally, when we used a series of models that account for socioeconomic, dietary, or treatment covariates; the association between race, ethnicity, and the microbiota was retained (Table 2). 

## 4. Discussion

Our pilot study shows that it is feasible to recruit a diverse sample to participate in studies of CRC and the gut microbiome. Our participation rate (58%) was higher than expected given the many barriers that our patients faced, including illness and pandemic-related restrictions [37]. The racial and ethnic diversity of the study team may have contributed to our recruitment efforts. Furthermore, prior to implementation, we sought input from thought leaders in the field of recruitment on appropriate telephone scripts to capture as diverse a population as possible. However, there was room for improvement in the feasibility and acceptability of the study procedures. Our study relied primarily on chart review of provider schedules which generated a large sampling frame, but from which many patients were excluded due to strict eligibility criteria. We also relied primarily on telephone calls to reach potentially eligible participants which led to high rates of attrition during the recruitment process. In part, this was due to the ongoing COVID-19 pandemic and restrictions on in-person research during the study period.

Once enrolled, most of our population was able to complete all components, and all returned stool samples were evaluable. Although only those who returned samples did exit interviews, they provided insight into reasons for enrolling and suggestions for improvement. One common issue was that the stool collection process was messy, and instructions could have been clearer. These are both relatively easy issues to address in the future, with the use of larger collection tubes and the incorporation of more patient-friendly collection guides and video (or in-person) demonstrations of sample collection. Several participants also noted that the diet survey was long, and that recall was difficult. Recall bias is certainly a consideration in the analysis of any dietary data, particularly as participants were asked to reflect on their diet for the past month. Simpler and patient-friendly methods of collection such as daily diaries (including photo journals) may address this issue of recall, but conversely, they may not provide the level of granularity necessary to quantify the intake of specific nutrients in relation to the gut microbiota. The type of diet instrument used should ideally strike a balance between minimizing the burden on the study participant but also maximizing the quality of data collected for the desired analysis. 

We demonstrated that we can successfully extract DNA and perform microbiota analyses from stool samples that relied on patient self-collection in a diverse cohort, setting the stage for larger studies utilizing similar methods. Our study was designed as a recruitment feasibility pilot, and, as such, it was not powered a priori to detect associations between the microbiome and diet, lifestyle, or sociodemographic variables. Despite this limitation, we detected an association between race and ethnicity and the microbiota that was retained even when accounting for potential confounders (prior treatment variables, diet, insurance, and education). Specifically, we found that the Fusobacteriota phylum was more highly abundant in Black/African American individuals, which could contribute to increased CRC development and progression in this group [14,16,17,38]. Interestingly, we also found that prior ileostomy was associated with differences in microbial community structure, in line with prior results from a study of colorectal cancer survivors [36]. Although race and ethnicity-associated differences in the gut microbiome could not be explained by measured treatment, socioeconomic, or dietary covariates, future studies may help untangle the nuances of how treatment and diet impact the microbiota of cancer patients by combining a higher sample size with measurement of detailed surgical history and dietary intake, among other covariates. 

The main limitation of our study is its small sample size. Recruitment was limited by strict eligibility criteria and reliance on the telephone as the primary mode of contact. Although we sought to recruit equal numbers by race and ethnicity, we were limited by our catchment area, demonstrating the necessity to partner with other clinical sites to ensure the recruitment of diverse populations in the future. In-person recruitment, as well as modifications to remote recruitment, such as the use of voicemail, calling during business hours, flyers, emails, social media, and other direct-to-patient communication methods could be utilized to increase yield in larger studies. In the future, recruitment and consent should occur, if possible, during the same visit/phone call. 

Although our study excluded patients who had received prior antibiotics, it did include those who had received prior cancer-directed therapy, which may have longer-term effects on the gut microbiome. The primary objective of our pilot was to evaluate the feasibility of recruitment of diverse patients (by race, ethnicity, socioeconomics, and treatment history), and as discussed above, it was not powered a priori to detect microbiome differences by race. As such, we chose to include patients who had received prior therapy. Future studies could modify inclusion/exclusion criteria based on the primary objective. To test the effect of prior therapies on the microbiota, future studies could consider including all patients and stratifying by treatment or antibiotic use. Alternatively, to avoid confounding by prior treatment, studies could restrict the patient population to treatment-naive, new diagnoses only. However, given potentially low CRC case rates at individual hospitals/clinics, strategic partnerships across multiple clinical sites will be essential in recruiting an adequate sample size for a well-powered study in this setting. 

Also, here, we have presented an analysis of the microbiome of English-fluent patients recruited solely from San Francisco, California, which may have biased findings. Future, larger studies must recruit participants from diverse language backgrounds (e.g., Spanish fluent participants who identify as Hispanic/Latino) and take into account a patient’s environmental and cultural context, and the impact of geography (in the US and globally) on the gut microbiome. Lastly, because the results of our exit interviews were limited to only those participants who completed all study procedures, suggestions and feedback from these participants may not reflect all potential recruitment challenges. Future studies could include brief interviews of individuals who decline participation to provide a richer understanding of the acceptability of the study. 

## 5. Conclusions

To date, the role of the gut microbiome in CRC disparities remains poorly understood. Our pilot study serves as a first step in further exploring these disparities by establishing the feasibility of home stool microbiome collections in a diverse population of individuals recruited from two clinical sites. Despite the small sample size, the study revealed significant differences in the microbial diversity and composition among Black/African American, Hispanic/Latino, and non-Hispanic White, individuals with colorectal cancer. These findings highlight the need to conduct larger longitudinal studies to build on these findings and dissect the mechanisms driving changes in microbial communities and their effects on CRC risk. By understanding factors that shape the gut microbiota and the functional consequences in CRC patients, we may be able to develop data-driven strategies to manipulate the microbiota to narrow the gap in the racial and ethnic disparities in CRC incidence and mortality.

## Figures and Tables

**Figure 1 cancers-15-04546-f001:**
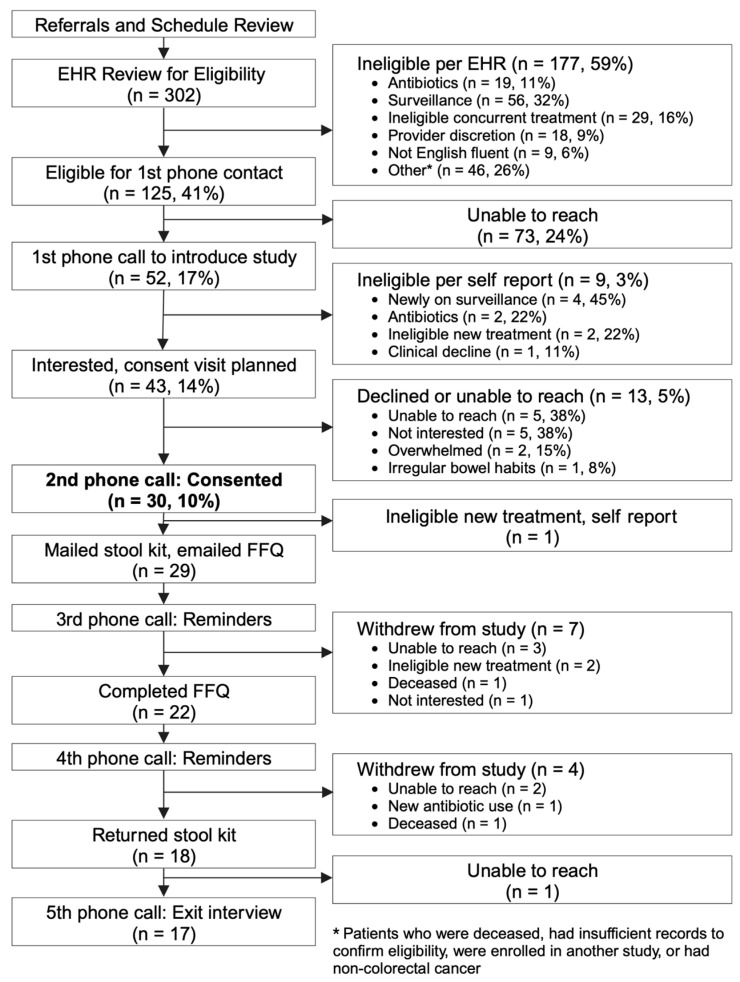
Recruitment and completion of study procedures.

**Figure 2 cancers-15-04546-f002:**
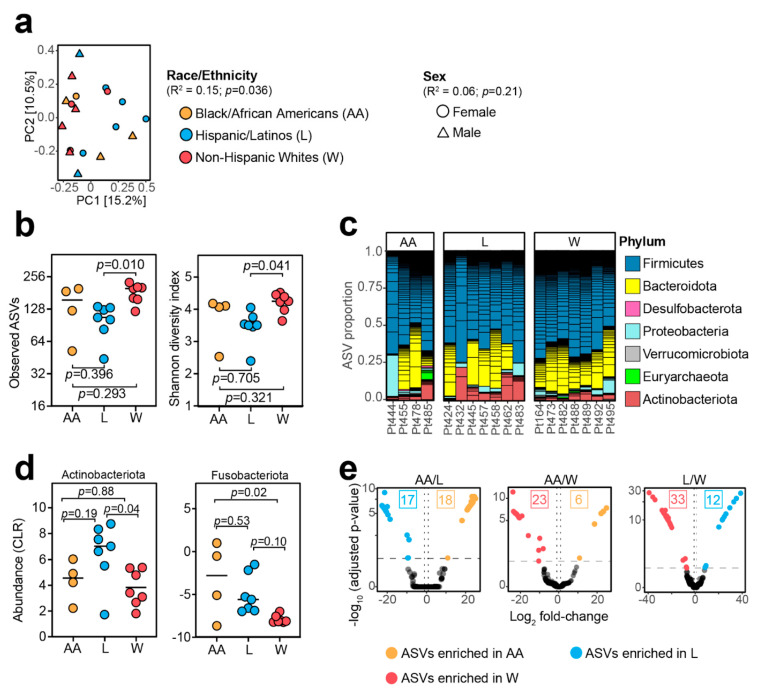
Race and ethnicity are associated with the gut microbiota of colorectal cancer (CRC) patients. (**a**) Principal coordinate analysis using Bray–Curtis distance matrix (permutational multivariate analysis of variance test using ADONIS statistical package). (**b**) Alpha diversity measurements using observed amplicon sequence variants (ASVs) and Shannon diversity index metrics (Appendix A). (**c**) Microbial community composition at the phylum level. Each bar represents stool from one participant. Short horizontal lines within bars represent different amplicon sequence variants. (**d**) Centered log-ratio (CLR)-transformed abundances of bacterial phyla with significant differences between groups (Appendix A). (**e**) Volcano plots of differential abundant ASVs from pairwise comparisons among Black/African American (AA), Hispanic/Latino (L), and non-Hispanic White (W) using DESeq package (see Appendix A for genus and species identification). Significant ASVs were defined as |log_2_ fold-change| > 1 and false discovery rate < 0.1. The number of significant ASVs is indicated in the plot. *p*-values shown in panels (**b**,**d**) are from one-way ANOVA tests with Tukey’s multiple comparison correction.

**Table 1 cancers-15-04546-t001:** Sample characteristics.

	N (%)
Enrolled Patients (*n* = 30)	
Recruitment site	
UCSF	26 (87%)
ZSFGH	4 (13%)
Race and Ethnicity	
Black/African American	5 (17%)
Hispanic	11 (37%)
Non-Hispanic White	14 (46%)
Age at diagnosis (yrs)	
≥50	12 (40%)
<50	18 (60%)
Cancer stage	
I and II	8 (27%)
III	9 (30%)
IV	13 (43%)
% Female	15 (50%)
% Male	15 (50%)
Anatomic Subsite	
Transverse	2 (7%)
Ascending/Cecum	6 (20%)
Sigmoid/Descending	5 (16%)
Rectosigmoid	2 (7%)
Rectum	15 (50%)
Survey and Stool Completion (*n* = 18)	
Race and Ethnicity	
Black/African American	4 (22%)
Hispanic/Latino	7 (39%)
Non-Hispanic White	7 (39%)
Insurance	
Private	8 (44%)
Medicare	3 (17%)
Medicaid	6 (33%)
Not reported	1 (6%)
Education	
High school	3 (17%)
Some college	6 (33%)
Trade/vocational	1 (6%)
Associate’s degree	3 (17%)
Bachelor’s	3 (17%)
Doctoral	2 (11%)
Prior Surgery to Resect Primary Tumor	
Yes	8 (44%)
No	10 (56%)
Prior Radiation	
Yes	4 (22%)
No	14 (78%)
Any Ostomy at Time of Collection	
Yes	8 (44%) (any colostomy or ileostomy)
No	10 (56%)
Ileostomy at Time of Collection	
Yes	2 (11%)
No	16 (89%)

**Table 2 cancers-15-04546-t002:** Race/ethnicity is associated with microbial composition even after accounting for patient covariates (Bray–Curtis ordination, term-wise PERMANOVA).

Category	Model	Race/Ethnicity R²	Race/Ethnicity *p*-Value
-	Distance ~ Race and Ethnicity	0.15	0.036
SES	Distance ~ Insurance + Education + Race and Ethnicity	0.14	0.073
Diet	Distance ~ BMI + FruitVeg + WholeGrain + Alcohol + Meat + Race and Ethnicity	0.17	0.029
Treatment	Distance ~ Ileostomy + Any ostomy + Radiation + Any surgery + Race and Ethnicity	0.14	0.055

## Data Availability

The data presented in this study are available on request from the corresponding author. The data are not publicly available due to patient confidentiality.

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
