# Peer review of "Associations between the Gut Microbiota, Race, and Ethnicity of Patients with Colorectal Cancer: A Pilot and Feasibility Study"

_cancers, 2023, doi:10.3390/cancers15184546_

Round 1
Reviewer 1 Report
General points:
The authors present a pilot study aiming to address whether there are any racially-associated microbial differences in colorectal cancer patients. This is timely, as the role of the microbiome in CRC development is being increasingly recognised, and because there are established racial differences in the incidence and treatment success of this disease.
The authors detail their methods fully, enabling other researchers to follow this work. Results are analysed and displayed using standard methods, appropriate for this type of data.
The main focus on the results and the discussion is to show the feasibility of this work. As such, there is not much novelty. The authors largely use protocols and reagents, such as the stool collection kits, designed for other studies. The novelty, and feasibility is therefore to show that this group of researchers, in these institutions, can carry out this sort of work in these populations. There is nothing wrong with this. It is often useful if this was to develop into a large project to have a publication with all the methods in one place. It does however put the importance of the work into context.
Specific points:
There was a lot of effort taken to conduct diet surveys, but no results presented, beyond how easy the patients found it. The same goes for socioeconomic measures.
Where were these patients in their treatment journeys. If they had undergone any surgical resection of part of their bowel, it is likely to have had a much greater influence on their microbiome than a lot of other factors. If one of the overall aims of this work is to look at risks, then pre-surgical samples might be most appropriate.
I appreciate that the authors have presented this as a feasibility study and pilot, and have not tried to over-analyse small samples numbers. However, they have presented some racially-associated microbial changes as significant. Is it feasibly to add dietary, social-economic and treatment (surgery yes/no being the main one) to the adonis analysis as factors to analyse before race. It might be that most of the association with race is actually something else like diet. It would also add to the strength of the pilot, showing how the authors will attempt to interpret their results in the light of lifestyle confounders which themselves are associated with race, but are known to affect the microbiome. How do they propose to quantify dietary data and socio-economic background in such a way as to make meaningful analysis possible.
Similarly, whilst it is perhaps beyond the scope of a pilot, it would be worth briefly discussing how this data could be analysed in the context of datasets from different countries. Non-hispanic white populations in Europe, hispanic populations in Central and South America, people of African descent in the Caribbean, Europe, or Africa. How much of the variation seen is in the context of people living in the area of this study, how much is cultural, and how much is genuine biological racial difference.
The font on figure 1 is quite difficult to read without quite a lot of zooming in.
Reviewer 2 Report
The paper, as a report on a feasibility study, fulfills all the necessary criteria of the publication. The study well describes the study design and the implementation, including the problems and difficulties. The authors and the readers can draw important conclusions and receive help for designing and performing a similar, but larger study.
Even certain conclusions related to the studied small sample may be important, primarily the data which underlines the ethnic differences regarding the composition of microbiota in CRC patients. Based on the small number of participants, however, these results need a confirmation and deeper analysis in larger studies.
There is, however, an issue, which has not been discussed in the paper. While the authors mentioned stratification as a possibility to increase the number of participants (instead of exclusion of those who received current antibiotic treatment), this is not necessarily the best solution. Antibiotics, chemotherapeutics, etc. may even cause a strong and long.term effect in the composition of the microbiota, thus exclusion of these patients is preferred over inclusion and stratification. Furthermore, it would be a good idea to change the recruitment method, and to recruit patients before they start to receive the anticancer therapy. This might require a multicenter study, since the number of incident cases of CRC patients probably would not be high enough in one clinic or hospital. These thoughts should be discussed in the article.
While the authors collected nutrtition-related data as well, an appropriate and convincing discussion regarding this data, and the potential effect of different dietary habits (as confounders) on the results is lacking. This should also be included in the discussion.
Round 2
Reviewer 1 Report
The authors have addressed my comments, and changed the manuscript in a satisfactory manner.